# Morphological and Biochemical Changes in the Mediterranean Cereal Cyst Nematode (*Heterodera latipons*) during Diapause

**DOI:** 10.3390/pathogens13080656

**Published:** 2024-08-02

**Authors:** Motasem Abumuslem, Halil Toktay, Monther T. Sadder, Abdelfattah A. Dababat, Nida’ M. Salem, Luma AL-Banna

**Affiliations:** 1Department of Plant Protection, School of Agriculture, The University of Jordan, Amman 11942, Jordan; motasemabumuslem@gmail.com (M.A.); n.salem@ju.edu.jo (N.M.S.); 2Department of Plant Production and Technologies, Faculty of Agricultural Sciences and Technologies, Niğde Ömer Halisdemir University, Nigde 51240, Türkiye; h.toktay@ohu.edu.tr; 3Department of Horticulture and Crop Science, School of Agriculture, The University of Jordan, Amman 11942, Jordan; sadderm@ju.edu.jo; 4International Maize and Wheat Improvement Centre (CIMMYT), Ankara 06000, Türkiye

**Keywords:** enzyme, genes, metabolism, molecular characterization, morphometrics, trehalase

## Abstract

The cereal cyst nematode (*Heterodera latipons*) is becoming an economically important species in global cereal production as it is being identified in many new cereal cultivated areas and causes significant losses. Consequently, understanding its biology becomes crucial for researchers in identifying its vulnerabilities and implementing effective control measures. In the current study, different morphological and biochemical changes of *H. latipons* cysts containing eggs with infective juveniles from a barley field in Jordan were studied during the summer of 2021, at two sample dates. The first, at the harvest of the cereal crop (June 2021), when the infective second-stage juveniles (J2s) were initiating diapause, and the second, before planting the sequent cereal crop (late October 2021), when the J2s were ending diapause. The studied population was characterized morphologically and molecularly, showing 98.4% molecular similarity to both JOD from Jordan and Syrian “300” isolates of *H. latipons.* The obtained results and observations revealed that there were dramatic changes in all the investigated features of the cysts and eggs they contained. Morphological changes such as cyst color, sub-crystalline layer, and thickness of the rigid eggshell wall were observed. A slight change in the emergence time of J2s from cysts was observed without any difference in the number of emerged J2s. The results of biochemical changes showed that the total contents of carbohydrates, glycogen, trehalose, glycerol, and protein were higher in cysts collected in October when compared to those cysts collected in June. The SDS-PAGE pattern indicated the presence of a protein with the size of ca. 100 kDa in both sampling dates, whereas another protein (ca. 20 kDa) was present only in the cysts of October. Furthermore, the expression of trehalase (tre) gene was detected only in *H. latipons* collected in October. The outcomes of this study provide new helpful information that elucidates diapause in *H. latipons* and may be used for the implementation of new management strategies of cyst nematodes.

## 1. Introduction

The Mediterranean cereal cyst nematode (MCCN; *Heterodera latipons*) belongs to the *Avenae* group of the genus *Heterodera* and is considered one of the most severely destructive nematodes on cereal crops [1]. *Heterodera latipons* was studied and described by Franklin [2], and up to now it has been reported in more than twenty countries across the continents according to the European and Mediterranean Plant Protection Organization [3]. MCCN was found to reduce the yield of barley by 50% in Cyprus, 20% and 30% in barley and wheat in Syria, respectively [4,5]. In Jordan, *H. latipons* was reported from several phytogeographical zones causing moderate to severe yield losses with 100% incidence in the Northern region [6,7,8].

*Heterodera latipons* is an obligatory sedentary endoparasite, completing only one generation per growing season [7,9,10]. Its life cycle is initiated by second-stage juveniles (J2), which are motile in the soil and penetrate the roots of a new cereal crop, where they become sedentary and develop into adult obese females after establishing permanent feeding sites. Obese females produce eggs that are retained inside their body, and at the end of their lifecycle, their cuticle hardens to form tough protective cysts, which remain attached to the roots or drop in the soil. The eggs complete embryonic development and contain newly molted J2s ready to hatch. The diapause of unhatched J2s is initiated by endogenous factors and terminated by exogenous stimuli [1]. The phenomenon of diapause was also reported in *H. avenae* [11,12,13,14,15,16], *Globodera rostochiensis* [17], and *G. pallida* [18].

Metabolic changes play an essential role in the survival of nematodes. A few studies have been conducted on the metabolic activities of parasitic nematodes during diapause as well as on its mechanism [19,20,21].

The molecular pathways (signaling pathways) of diapause in *H. latipons* are still unknown. In contrast, genetic and molecular studies on the model nematode *Caenorhabditis elegans* revealed a wealth of information for the research on other nematodes, where the signaling pathway of the dauer larva was well investigated [22]. Hochbaum et al. [23] found that under unfavorable environmental conditions the dauer diapause stage was controlled by endocrine signals. A population of a cereal cyst nematode, tentatively identified as *H. latipons*, was collected from a barley field in Jordan and used in this study. The correct identification of this population was necessary because several species of cyst-forming nematodes infest cereals in the Mediterranean basin. Their morphological characters are variable, making their identification challenging. Therefore, the objectives of the present study were as follows: (i) characterize morphologically and molecularly the population of the cereal cyst nematode tentatively identified as *H. latipons*; from Jordan; (ii) determine the morphological and biochemical changes of cysts and enclosed J2s in eggs of this population during their diapause, and (iii) investigate whether genes responsible for diapause in *C. elegans* would have orthologs in this putative population of *H. latipons*.

## 2. Materials and Methods

### 2.1. Nematode Sampling, Species Identification, and Emergence Assay

A barley field naturally infested by *H. latipons* located in Madaba, Jordan (31.709376 N, 35.818771 E), was used and soil samples were collected for cyst extraction. Random composite samples were collected at two different times during the natural diapause of *H. latipons*: early June 2021, after the barley harvest time, and late October 2021, prior to the sowing time of barley or wheat crops. The maximum temperature in Madaba during June, after harvesting barley, ranges from 24 to 38 °C. It then increases in the following months of July, August, and September, before slightly decreasing in October, with a range of 25 to 34 °C (Appendix A). It is worth mentioning that no rain events occurred from June to October 2021. The relative humidity in June, July, August, September, and October was marginally low, averaging 44%, 43%, 44%, 53%, and 51%, respectively (Appendix A).

Cysts of *H. latipons* were extracted from the collected soil using the cyst flotation method as per Jones [24]. Extracted cysts were cleaned by rinsing them several times in sterile distilled water (SDW) before being used in the assays.

For an accurate identification of the population from Jordan, comparisons of the morphological features of the conical portion of the posterior body of the cysts and morphometrics of 10 J2s and cysts were made with those of the described populations of *H. latipons* and other species in the group *avenae* [2,25]. In addition, the findings of the morphological analysis were combined with those of the molecular characters. The obtained morphometrical data of J2s and cysts were run through discriminant analysis (DA) using IBM SPSS statistics software (version 18.0) and compared with the original description of *H. latipons*, the different Jordanian populations of *H. latipons*, as well as other *Heterodera* species including *H. avenae* and *H. schachtii* [2,7,8,25]. To confirm the results of the morphological analysis, molecular analyses of the cysts of the putative *H. latipons* population and other populations of this species from GenBank were conducted using the 28S rDNA gene sequences. The D2 and D3 expansion segments of the 28S rDNA were amplified using the primer set D2A-F (5′-ACAAGTACCGTGAGGGAAAGTTG-3′) and D3B-R (5′-TCGGAAGGAACCAGCTACTA-3′) [26]. Total DNA was extracted from the cyst of *H. latipons* following the procedure described by Subbotin et al. [26]. The PCR product (749 bp) was purified and bidirectionally sequenced (Macrogen, Seoul, Republic of Korea). The obtained sequences were compared with those sequences available in the GenBank using BLASTn function [27].

Low temperatures, 8–10 °C, induced J2s emergence in studies conducted by Scholz and Sikora [10]. The effect of these exogenous stimuli on J2s emergence was tested in the cysts collected at the two sampling dates. For this test, three replicates of 10 cysts each were placed into a 1.5 mL tube filled with 100 μL of SDW, and then maintained at 8–10 °C in an incubator to verify the emergence of J2s from the cysts.

To conduct a morphological comparison (physical and developmental) between the two sampling dates, 10 cysts were randomly collected from each studied date. The comparison was based on various factors including cyst color, sizes (length [L] and width [W]), mean number of eggs/cyst, egg stage, and the presence of a sub-crystalline layer (an external layer that surrounds the cuticle). Cyst volume was determined using the ellipsoidal volume equation [4/3π × (L/2) × (W/2)^2^], and *t*-test was performed at *p* = 0.05.

### 2.2. Quantification of Biochemical Indices during the Diapause of H. latipons

The total contents of carbohydrates, glycogen, and trehalose were determined based on the anthrone–sulfuric acid method [21,28,29], and the colorimetric response was compared to a standard curve based on glucose and expressed as ng per cyst.

Glycerol content was measured according to the method of Wu and Yuan [30]. The total content of soluble protein was determined based on the Bradford method [28], and the colorimetric response was compared to a standard curve based on bovine serum albumin (BSA) and expressed as ng per cyst. Three replicates were used for each quantification assay and the t-test was performed at *p* = 0.05.

### 2.3. Qualitative Analysis of Protein during the Diapause of H. latipons

To detect differences in total protein bands between the two sampling dates after electrophoresis, a modified SDS-PAGE assay [21] was conducted with three replicates. Thirty washed cysts of *H. latipons* from each sampling date were transferred into a 1.5 mL tube containing 40 μL of extraction buffer (0.03 M, pH 6.8, Tris-HCl, 2% (*v*/*v*) Triton X-100, 6% (*v*/*v*) glycerol, 0.01 M NaCl) and 2 μL of 50× protease inhibitor cocktail (Promega, Madison, WI, USA). The cysts were ground thoroughly on ice for 3 min using a pellet pestle motor and then incubated on ice for 40 min with flecking each 10 min. The homogenate was centrifuged for 10 min at 14,000 rpm at 4 °C, the supernatant was kept at −20 °C until required.

Fifteen µL of extracted protein was mixed with 2× SDS-PAGE sample buffer (0.5 M, pH 6.8, Tris-HCl, 50% glycerol, 1% bromophenol blue, 10% SDS, 5% β-mercaptoethanol) and then separated using a 10% (*w*/*v*) resolving gel and a 5% (*w*/*v*) stacking gel. The electrophoresis was conducted at 100 V through the stacking gel and 120 V through the resolving gel, using 1% Tris/glycine buffer (Fisher, Waltham, MA, USA). The running was stopped once the indicator reached a distance of 1.0 cm from the bottom of the gel. The gel was stained by Coomassie brilliant blue R-250, then de-stained in a solution of methanol:acetic acid:water (10:10:80, v:v:v) until the protein band became visible.

### 2.4. Identification of C. elegans Diapause Related Gene Homologues Available in H. latipons

Based on published literature [22,31], twenty candidate genes related to diapause in *C. elegans* were selected and they included twelve abnormal dauer formation (*daf*) genes, five alpha, alpha-trehalose glucohydrolase (*tre*) genes, two trehalose-6-phosphate synthase (*tps*) genes, and a phosphatidylinositol 3-kinase age-1 (*age*-1) gene. These genes were used to find available orthologs among plant parasitic nematodes (PPNs). The sequences of these twenty genes were analyzed under BLASTx function [27] with a selection option (organism) of nematode namely as follows: *Aphelenchus avenae*, *Bursaphelenchus okinawaensis*, *B. xylophilus*, *H. glycines*, *Meloidogyne enterolobii*, and *M. graminicola*. Consequently, ten orthologs from PPNs were detected and retrieved (Appendix A). To find orthologs in *H. latipons*, RNA-Seq data of the related species *H. avenae* were used as only very limited GenBank entries are available for *H. latipons*. RNA-Seq data for the related *H. avenae* were retrieved from the available Sequence Read Archive (SRA) in GenBank [27]. The RNA-Seq data were assembled using the Qiagen CLC Genomics Workbench version 9.0.1 software (Qiagen, Venlo, The Netherlands). BLASTx results from PPNs were blasted using assembled *H. avenae* contigs using the BioEdit version 7.0.5.3 software [32], and best hits (according to E-value) were collected from the assembled contigs (Appendix A).

#### 2.4.1. RNA Extraction, Primer Design and PCR

The total RNA of fifteen egg-filled cysts from each sampling date was extracted using a RNeasy^®^ Plus Micro Kit (Qiagen, Hilden, Germany) following the manufacturer’s instructions. DNA digestion was conducted on the column during RNA extraction using RNase-Free DNase set (Qiagen, Germany) as recommended. Total RNA was checked for its quality and quantity using a NanoDrop One^C^ (Thermo Scientific, Waltham, MA, USA). The first strand of complementary DNA (cDNA) was synthesized using the AccuPower^®^ RT PreMix kit (Bioneer, Daejeon, Republic of Korea) following the manufacturer’s instructions. Primer design was conducted online using the Primer3Plus program [33] for the ten specific genes retrieved from the *H. avenae* assembly (Appendix A).

Three μL of cDNA from each sampling date was transferred to a 0.2 mL PCR tube containing 7.5 μL 2× TransTaq^®^-T PCR SuperMix (+dye), 2 μL from each forward and reverse primers (10 μM) (Appendix A), and nuclease-free water to a final volume of 15 μL. The PCR program consisted of one cycle of denaturing at 95 °C for 5 min, followed by thirty-five cycles of 30 s denaturing at 95 °C, 30 s annealing at 50 °C, 1 min extension at 72 °C, and finishing with a final extension of 10 min at 72 °C. The amplified products were separated by 1.8% agarose gel electrophoresis at 100 V for 35 min, then checked by the gel documentation system (BIORAD Gel doc 2000, Hercules, CA, USA). Each sampling date involved the utilization of three replicates for every primer pair.

#### 2.4.2. Sequencing and Data Analysis

The target bands were excised from agarose gel and cleaned up using Zymoclean Gel DNA Recovery Kit (Zymo Research, Orange, CA, USA) according to the manufacturer’s instructions. The purified DNA was sent for direct sequencing of both directions at a sequencer facility (Macrogen, Seoul, Republic of Korea). The obtained sequences of each gene both forward and reverse were analyzed using the BLASTx and BLASTn functions [27]. Multiple alignments were performed for trehalase of *H. latipons* and homologous proteins from different nematodes using ClustalW function available in BioEdit version 7.0.5.3 software. The functional domain was detected using the database of protein families (Pfam version 35.0).

## 3. Results

### 3.1. Identification of Madaba Jordanian isolate of H. latipons

The morphological identification of the cereal cyst nematode population from Jordan as *H. latipons* was based on the comparisons of the morphological features and morphometrics of its J2s and cysts with those of *H. latipons* in the original description and other described populations [2] (Table 1 and Table 2) (Appendix A). In addition, DNA sequence comparisons revealed best hit sequences with 98.4% similarities to both *H. latipons* JOD isolate (Accession HM560854) and *H. latipons* Syrian “300” isolate (Accession DQ328687).

The emergence of J2s from *H. latipons* cyst was observed in June and October samples after 39 and 32 days of incubation, respectively. For the cysts collected in June, the mean emergence percentage and total number of emerged J2 were 31% and 600, respectively, when compared to cysts collected in October, which had a mean emergence percentage of 32% and a total number of emerged J2 of 613.

The canonical discriminant analysis was carried out using the standardized canonical discriminant function coefficients for 23 morphometrical traits showing that the first four functions accounted for 94% of the total variation (Table 3). The first canonical discriminant function accounted for 61.4% and was strongly influenced by the distance of vulval slit to semi-fenestra, tail length, the distance of anterior end to end of the median bulb, and the b^m^ = body length/distance from lip to end of the median bulb (Appendix A). The second function accounted for 14.3% and was influenced by head height, tail length, and body length. The third function accounted for 9.6% and was influenced by fenestral width, the distance of anterior end to end of the median bulb, and a = L/W (body length/midbody width). The fourth function accounted for 8.7% and was influenced by the tail length and body width at the anus. The discriminant analysis (Figure 1) showed a clear separation of *H. latipons* populations including one of this study from other *Heterodera* species including *H. avenae* and *H. schachtii*.

Comparing the morphology of cysts collected on the two different sampling dates, it was observed that the June-recovered cysts exhibited a 60% light brown/tan color and 40% dark brown color, whereas the October-recovered cysts exhibited an 80% dark brown color and 20% light brown/tan color. The cysts recovered in June showed a slight reduction in size compared to the October-recovered cysts. The length, width, and overall cyst volume of the June-recovered cysts were measured to be 607 ± 59.4 µm, 466 ± 56.6 µm, and 69 × 106 μm^3^, respectively, while the corresponding measurements for the October-recovered cysts were 615 ± 54.3 µm, 502 ± 69.3 µm, and 81 × 106 μm^3^, respectively. Statistical analysis revealed no significant difference between the two sets of cysts (*p* > 0.05). The mean number of eggs/cyst in the June-recovered cysts was 65 ± 7, while in the October-recovered cysts it was 70 ± 10. Regarding the egg stage, it was observed that both the cysts recovered in June and October contained unhatched J2s. In the majority of June-recovered cysts, the sub-crystalline layer was present and firmly attached to the cyst body, whereas it was generally absent in the October-recovered cysts (Figure 2). Furthermore, it was observed that the thickness of the rigid eggshell wall in the October-recovered cysts was higher compared to the June-recovered cysts (Figure 2).

### 3.2. Total Content of Carbohydrate, Glycogen, Trehalose, Glycerol, and Protein of H. latipons during Diapause

The changes in biochemical indices of *H. latipons* during diapause indicated that the total content of carbohydrates, glycogen, trehalose, glycerol, and protein in the cysts of June was 242 ± 7, 41 ± 3, 188 ± 35, 56 ± 45, and 215 ± 26 ng per cyst, when compared to cysts collected in October 471 ± 29, 77 ± 19, 325 ± 67, 87 ± 53, and 357 ± 22 ng per cyst, respectively (Figure 3). The content of carbohydrates, trehalose, and protein in the cysts of October was significantly higher than that in the cysts of June. There were no significant differences in the total glycogen and glycerol contents between the cysts of the two collection dates.

### 3.3. SDS-PAGE Analysis of Protein

The SDS-PAGE pattern indicated the presence of a protein with the size of ca. 100 kDa in the cysts of both June and October sampling dates (Figure 4, blue arrows), whereas another protein (ca. 20 kDa) was present in cysts of October but absent in the cysts of June (Figure 4, black arrows).

### 3.4. Genetic Regulation of Arrested Development in H. latipons

#### 3.4.1. PCR Result

Based on the hypothesis that arrested development in *H. latipons* could be genetically regulated via orthologs of the *daf*-genes, *tre*-genes, *tps*-genes, or *age*-1 gene of *C. elegans*, designed primers were applied in PCR utilizing *H. latipons* expressed sequence tags (ESTs) from the two sampling dates (early June and late October 2021).

Analyzing PCR products of the June 2021 sampling date revealed no fragment. In contrast, the analysis of PCR products for the October 2021 sampling date revealed a successful amplification of four primer combinations: (i) H.l-F05 and H.l-R05, (ii) H.l-F08 and H.l-R08, (iii) H.l-F09 and H.l-R09, and (iv) H.l-F10 and H.l-R10, a single band amplified of the expected fragment size (Appendix A). Three primer pairs (i) H.l-F02 and H.l-R02, (ii) H.l-F03, H.l-R03, and (iii) H.l-F04 and H.l-R04 resulted in unexpected fragments and the remaining primer pairs did not show any fragments (Appendix A).

#### 3.4.2. Sequence Analysis

A total of seven different amplification products from the October 2021 sampling date were analyzed by sequencing. Based on BLASTx and BLASTn functions [27] the obtained sequences were compared with those sequences deposited in the GenBank and showed identities and similarities in some cases, as listed in Appendix A.

However, the BLASTx analysis in the NCBI revealed that only *tre*-5 gene showed identities with the trehalase protein sequences deposited in the GenBank (Table 4 and Appendix A). The best hit sequences for the trehalase protein showed a total of 57% and 42% identities with the trehalase protein of *Meloidogyne enterolobii* (Accession CAD2205081.1) and *Globodera pallida* (Accession KAI3418038.1), respectively. On the other hand, no significant similarity was found from the other obtained sequences when compared to those sequences available in the GenBank. The sequence of the *tre* gene was deposited in the GenBank database under the accession number OP150195.

#### 3.4.3. Homology Analysis of Trehalase Protein

The ClustalW multiple alignment analysis revealed that the trehalase protein sequence of *H. latipons* was found to cover around 100 amino acids homologous from other nematodes, as shown in Figure 5. Interestingly, the lysine-rich motif (KKQNSKLVRQETIKISG) highlighted within the red box in Figure 5 was only found in the trehalase protein sequence of *H. latipons*. In contrast, this particular motif was identified in three animal parasitic nematodes, namely *Ancylostoma duodenal* (Accession KIH45727.1), *Oesophagostomum dentatum* (Accession KHJ92105.1), and *Wuchereria bancrofti* (Accession VDM19656.1). This finding distinguishes the lysine-rich motif as unique to the *H. latipons* sequence when compared to the homologous proteins of PPNs. Furthermore, the functional analysis of the obtained protein from *H. latipons* indicated the presence of Pfam domain which is conserved among trehalase proteins from different nematode species.

## 4. Discussion

Nematodes have different strategies to face different challenges in their life cycles; one of these strategies is diapause. The diapause of MCCN eggs in Mediterranean climates is obligate and durable and it is initiated by endogenous factor and terminated by resaving exogenous stimulus and occurs when the climate is hot and dry and ends when the temperature of soil becomes low, and the moisture increases [1,34].

Morphology and ultrastructure of the nematodes are essential to understand physiological functions. Egg-filled cyst and rigid eggshell wall afford protection to the unhatched J2s. The thickness of the adult cuticle of *C. elegans* increases with aging [35]. Likewise, the rigid eggshell wall of *H. latipons* in the present study was thicker in the cysts of October prior to the next growing season than those in the cysts of June at the end of the growing season (Figure 2). This might help the J2s to remain viable for a long time by decreasing the direct effect of hostile environmental conditions.

Our findings indicated that the emergence percentage and the number of emerged J2s were similar at both sampling dates, even though J2s from October cysts began emerging seven days earlier than those from June cysts. In contrast, the study by Scholz and Sikora [10] demonstrated that four-month-old cysts had more emerged J2s compared to one-month-old cysts. However, it is important to note that in their experiment, cysts of all ages were prestored for five to seven months at 20 °C. In our study, cysts were extracted from the soil immediately after sampling and then subjected to the hatching assay.

The metabolic changes play an essential role in the survival of nematodes. The eggs of nematodes are primarily composed of carbohydrates, proteins, and lipids [36]. Lipids and carbohydrates serve as storage materials for survival and provide energy for nematodes [19,36]. The J2 inside the egg is surrounded by perivitelline fluid, which contains trehalose [37,38]. Trehalose is a disaccharide that has roles in the cold tolerance strategies of nematodes and protection against environmental stresses [39,40]. The trehalose has been used as a marker for the viable eggs in the potato cyst nematode [41,42].

In the present study, the total contents of carbohydrates, glycogen, and trehalose were higher in the cysts of October than those the cysts of June. These results may explain the requirements of the nematodes for more energy at this stage of their lives in order to reactivate the J2s and end the diapause (especially, the trehalose has a role in sugar transport) and eventually to start local exploration and emergence.

The link between glycogen and trehalose metabolism was observed [43]. For example, the entomopathogenic nematode *Steinernema feltiae* showed a twofold increase in trehalose and a reduction in glycogen accumulation after desiccation of the nematode [43]. Similarly, a twofold increase in trehalose and a reduction in glycogen accumulation was observed in *H. avenae* at the end of the growing season [28], while in our study we observed a twofold increase in both trehalose and glycogen.

Glycerol plays a crucial role in rapidly balancing the osmotic pressure in nematodes [44]. In the cysts of October, the total content of glycerol was higher than that in the cysts of June. Increasing the glycerol content prior to the next growing season might help the nematode to leak the trehalose and uptake water in order to reactivate the J2s.

Proteins are one of the components of eggs; the total content of protein in our study was higher in the cysts of October than those of June. The SDS-PAGE profile revealed that the cysts of October had an extra band than those of June. The presence of an extra protein band prior to the next growing season may signify the presence of a new protein/gene having a role in the diapause signaling pathways to end the diapause period.

Our results showed that the total amounts of carbohydrates, glycogens, trehalose, and proteins were lower in cysts of *H*. *latipons* regardless of the sampling date than those of *H*. *avenae* and the reason might be that the later cyst nematode is larger in size and weight [28,29]. On the other hand, *H. latipons* cysts collected in October have a higher amount of glycerol than those cysts of *H. avenae* although *H. latipons* is smaller in size and weight [28,29].

In the current study, the total content of carbohydrates, glycogen, trehalose, and glycerol was higher in the cysts that were collected in October 2021 (before the sowing of barley seeds, during the termination of diapause) than those collected in June 2021 (after harvesting of barley). Similarly, Mo et al. [28] reported that brown cysts of *H*. *avenae* have more trehalose and glycerol than white cysts. However, brown cysts of *H*. *avenae* have lower contents of total carbohydrates, glycogen, and proteins when compared to white cysts of the same sampling date at the end of the growing season [28].

In the present study, different primers were designed in order to detect some of the genes that are responsible for the arrested development in the *C. elegans* and which could be orthologous in the *H. latipons* cyst nematode. The PCR tests of the June 2021 sampling date revealed no fragments, while in the October 2021 sampling date some fragments of certain genes were observed. The developmental stage of nematodes could play an essential role in the expression of the diapause genes. Also, conditions in October could have a positive impact on diapause-responsive genes.

On the other hand, the sequences analysis revealed that no identities to the respective *age*-1 gene, *daf*-genes, or *tps* gene of the *C. elegans* or any of the PPNs were observed. But the sequence of the *tre* gene showed identities to other trehalase sequences deposited in the GenBank. The reason that no identities with other sequences were found in the GenBank could be either that the searched genes are not present in the *H. latipons*, or the arrested development of *H. latipons* may be regulated by completely different genetic pathways. In fact, in the GenBank [27] the *age*-1 gene, *daf*-genes, as well as the *tps* gene are present in PPNs. Despite the existence of *age*-1, *daf*, as well as *tps* genes in some of the PPNs, their precise regulatory function remains unclear.

The metabolism of trehalose is catalyzed by different enzymes; one of these enzymes is the trehalase enzyme which is responsible for catalyzing the hydrolysis of sugar. Trehalase activity was detected in different nematodes [21,28,29,45]. In the present study, partial *tre* gene was detected in *H. latipons* by using a gene-specific primer pair. The presence of *tre* gene suggested that the enzyme has an important function in the physiology of nematodes. Further investigations are required to clarify the function of the lysine-rich motif in our novel sequence as well as to find the diapause regulatory genes in *H. latipons*. Using RNA interference/post-transcriptional gene silencing might help us to establish a new control strategy by knockdown of the gene expression.

## Figures and Tables

**Figure 1 pathogens-13-00656-f001:**
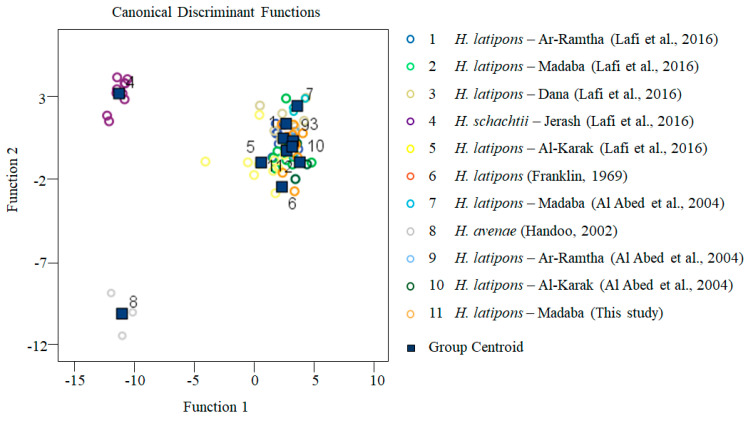
Canonical discriminant analysis for *Heterodera latipons* (this study) and different populations of *Heterodera* spp. based on 23 morphometrical characters of J2s and cysts. (n = 10 for J2s and cysts, n = 5 for vulval cone morphometrics) [2,7,8,25].

**Figure 2 pathogens-13-00656-f002:**
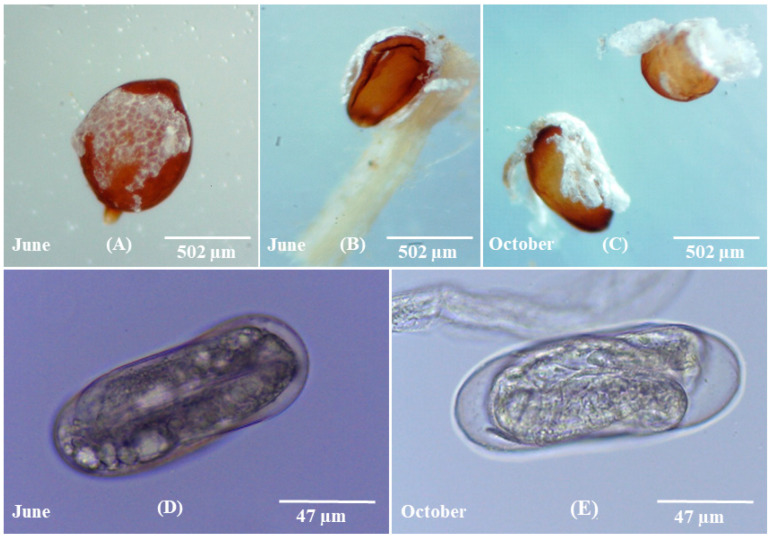
Morphological comparative studies of *Heterodera latipons* at the two sampling dates (early June and late October 2021) showing: (**A**–**C**) cysts; the sub-crystalline layer in the June recovered cysts was more robustly attached to the body of the cyst than those in the October recovered cysts. (**D**,**E**) Eggs; the rigid eggshell wall in the October cysts was thicker than those in the June cysts.

**Figure 3 pathogens-13-00656-f003:**
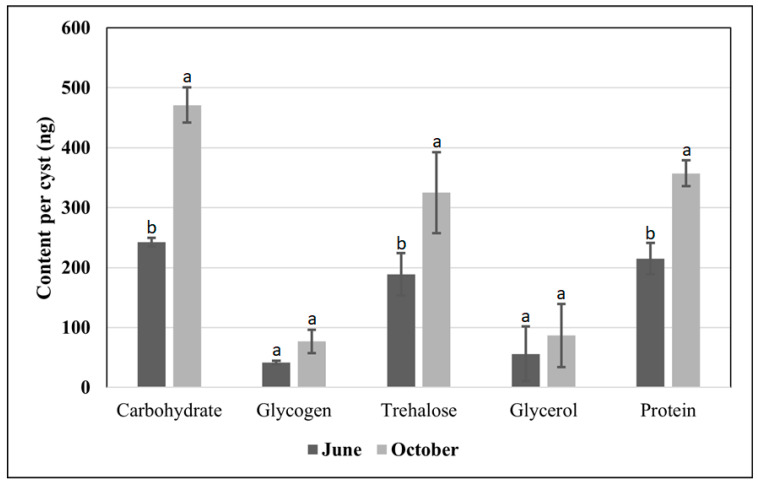
Total carbohydrate, glycogen, trehalose, glycerol, and protein content of *Heterodera latipons* cysts at the two sampling dates (early June and late October 2021) during its diapause. Error bars indicate ± SD; n = 3. Different letters indicate statistically significant differences (*p* ≤ 0.05).

**Figure 4 pathogens-13-00656-f004:**
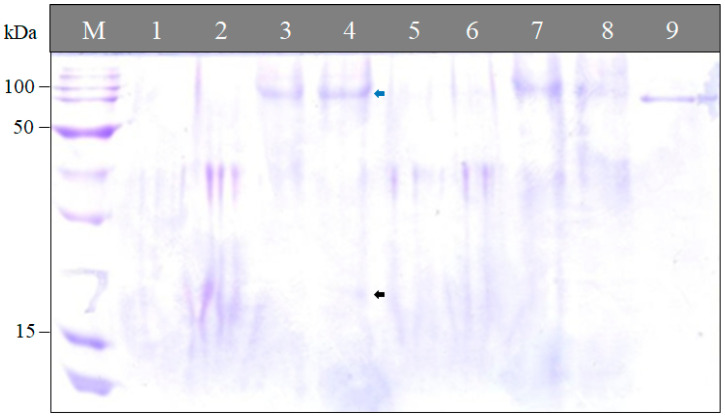
Protein patterns of *Heterodera latipons* cysts at the two sampling dates (early June and late October 2021) during its diapause. Lane M: protein ladder 0.1 µg/µL (Promega, Madison, WI, USA). Lanes 1, 3, 5 and 7: soluble protein of June cysts. Lanes 2, 4, 6 and 8: soluble protein of October cysts. Lane 9: bovine serum albumin (BSA).

**Figure 5 pathogens-13-00656-f005:**
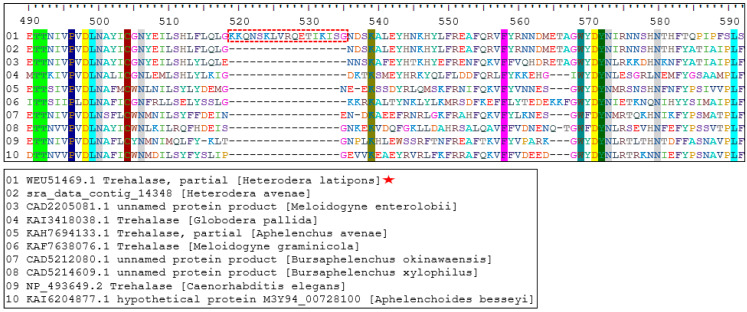
Alignment of the amino acid sequence of the trehalase protein of *Heterodera latipons* and homologous proteins from different nematodes by using ClustalW Multiple Alignment algorithm (BioEdit ver. 7.0.5.3). 
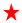
: Trehalase protein sequence of *H. latipons* Madaba Jordanian isolate.

**Table 1 pathogens-13-00656-t001:** Morphological and morphometric characteristics of J2s of *Heterodera latipons* Madaba Jordanian isolate.

No.	Characteristic (10 Individual J2s)	Measurements in µm	
		Ranges	Mean ± SD *
1	Body length (L)	420–560	473 ± 43.3
2	Midbody width (W)	19.03–22.49	21 ± 0.8
3	Head height	3.46–5.19	5 ± 0.8
4	Head width	8.65–10.38	10 ± 0.7
5	Stylet length	22.49–27.68	25 ± 1.6
6	a = L/W	20.23–27.58	23 ± 2.2
7	bm = L/distance from lip to end of median bulb	5.84–8.42	7 ± 0.9
8	Distance (anterior end to end of median bulb)	62.28–77.85	71 ± 6.1
9	Distance (dorsal gland duct opening to stylet base) DEGO	4.2–5.19	5 ± 0.4
10	Distance (anterior end to start of overlapping)	77.85–95.15	85 ± 5.3
11	Distance (anterior end to end of overlapping)	140.13–164.35	150 ± 7.3
12	Distance (anterior end to excretory pore)	81.31–112	91 ± 8.7
13	Tail length	38.06–53.63	48 ± 4.9
14	Hyaline length	24.22–29.41	27 ± 1.8
15	Body width at anus	13.84–17.3	16 ± 0.9
16	c = L/tail length	8.63–11.77	10 ± 1
17	Hyaline length/stylet length	0.93–1.23	1 ± 0.1
18	Tail length/body width at anus	2.44–3.53	3 ± 0.3

* SD: Standard deviation.

**Table 2 pathogens-13-00656-t002:** Morphological and morphometric characteristics of cysts and vulval cone structures of *Heterodera latipons* Madaba Jordanian isolate.

No.	Characteristic		Measurements in µm	
			Ranges	Mean ± SD *
	**(10 individual cysts)**			
1	Body	length (excluding neck)	540–702	615 ± 54.3
		width	396–567	502 ± 69.3
		neck	27–90	56 ± 22
	**(5 individual vulval cone)**			
2	Underbridge	length	95.15–138.4	121 ± 18.4
		width	9.51–15.57	13 ±2.57
3	Fenestral	length	60.55–88.23	71 ± 12.2
		width	18.16–22.49	20 ± 1.7
4	Semi-fenestral length (c)		12.11–17.3	12 ± 17.3
5	Vulval slit (s)		12.11–15.57	12 ± 15.6
6	Width of vulval bridge (b)		25.95–34.6	26 ± 34.6
7	Bullae (absent or present)		Few scattered to absent	

* SD: Standard deviation.

**Table 3 pathogens-13-00656-t003:** Eigen values and percentage of variability explained by each canonical discriminant function for *Heterodera latipons* (this study) and different populations of *Heterodera* spp. based on 23 morphometrical characters of J2s and cysts.

Function	Eigen Value	% of Variance	Cumulative %
**1**	31.712	61.4	61.4
**2**	7.381	14.3	75.7
**3**	4.965	9.6	85.4
**4**	4.471	8.7	94
**5**	1.635	3.2	97.2
**6**	0.818	1.6	98.8
**7**	0.270	0.5	99.3
**8**	0.179	0.3	99.6
**9**	0.103	0.2	99.8
**10**	0.079	0.2	100

**Table 4 pathogens-13-00656-t004:** Best alignment hits of BLASTx sequence of trehalase protein using the primer set H.l_F10 and H.l_R10 to detect the presence of *tre* gene in *Heterodera latipons*.

No.	Description [Organism]	E Value	Percent Identity	Accession
1	Unnamed protein product [*Meloidogyne enterolobii*]	1 × 10^−32^	57	CAD2205081.1
2	Trehalase [*Globodera pallida*]	1 × 10^−15^	42	KAI3418038.1
3	Trehalase [*Aphelenchus avenae*]	4 × 10^−15^	42	KAH7694133.1
4	Trehalase [*Meloidogyne graminicola*]	2 × 10^−13^	38	KAF7638076.1
5	Unnamed protein product [*Bursaphelenchus okinawaensis*]	2 × 10^−12^	44	CAD5212080.1
6	Unnamed protein product [*Bursaphelenchus xylophilus*]	2 × 10^−12^	42	CAD5214609.1
7	Trehalase [*Caenorhabditis elegans*]	3 × 10^−12^	43	NP_493649.2
8	Hypothetical protein M3Y94_00728100 [*Aphelenchoides besseyi*]	8 × 10^−12^	40	KAI6204877.1

## Data Availability

Data generated in this study are available upon reasonable request to the corresponding author.

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
