# Peer review of "Morphological and Biochemical Changes in the Mediterranean Cereal Cyst Nematode (Heterodera latipons) during Diapause"

_pathogens, 2024, doi:10.3390/pathogens13080656_

Round 1
Reviewer 1 Report
Comments and Suggestions for Authors
Review of “Morphological and biochemical changes in the Mediterranean cereal cyst nematode (Heterodera latipons) during diapause”
Abumuslem et al. explore morphological and biochemical aspects behind the obligate diapause of Heterodera latipons, a plant-parasitic nematode of economic relevance. The paper is well written, but I provide some comments to improve the current version of the manuscript.
Major comments:
Introduction: well-structured and objective approach to the state of the art.
Material & Methods: proper references to previously published methodology and methods seem adequate.
Results & Discussion: results, including tables and figures, are nicely presented. Logical interpretation of results and conclusions.
Specific comments:
Introduction
Line 43: remove “nematode”; when you refer the species name, adding “nematode” right after is redundant. I suggest changing this consistently throughout the manuscript
Line 45: factors
Line 46: stimuli; a reference is missing for this sentence. Perhaps authors can move ref #10 right after “stimuli” and slightly change the next sentence to “The phenomenon of diapause was also reported in H. avenae [11,12,13,14,15,16], Globodera rostochiensis [17], and G. pallida [18].”
Material and Methods
Line 127: remove “nematode” (it’s redundant)
Line 132: Bursaphelenchus okinawaensis is not a plant-parasitic nematode; it’s a mycetophagous (fungus-feeding) nematode
Results
Line 228: Scale bars are missing in Figure 2
Lines 248-249: Italicize Heterodera latipons; also, even though it may be obvious, the meaning of different letters and their statistical significance should be clearly stated in the legend.
Line 297: ClustalW Multiple Alignment algorithm, rather than program
Discussion
Line 376: is it really worth abbreviating RNAi and PTGS? This is the only line in the entire manuscript where they are referred…
Author Response
On behalf of all authors i would like to thank this reviewer for his/her valuable comments on the manuscript. We revised the comments/corrections as kindly indicated.
Reviewer no.1
Comments and Suggestions for Authors
Review of “Morphological and biochemical changes in the Mediterranean cereal cyst nematode (Heterodera latipons) during diapause”
Abumuslem et al. explore morphological and biochemical aspects behind the obligate diapause of Heterodera latipons, a plant-parasitic nematode of economic relevance. The paper is well written, but I provide some comments to improve the current version of the manuscript
Major comments:
Introduction: well-structured and objective approach to the state of the art.
Material & Methods: proper references to previously published methodology and methods seem adequate.
Results & Discussion: results, including tables and figures, are nicely presented. Logical interpretation of results and conclusions.
Specific comments:
Introduction
Line 43: remove “nematode”; when you refer to the species name, adding “nematode” right after is redundant. I suggest changing this consistently throughout the manuscript
- The required change has been addressed.
Line 45: factors
- The required change has been addressed.
Line 46: stimuli; a reference is missing for this sentence. Perhaps authors can move ref #10 right after “stimuli” and slightly change the next sentence to “The phenomenon of diapause was also reported in H. avenae [11,12,13,14,15,16], Globodera rostochiensis [17], and G. pallida [18].”
- The required change has been addressed.
Material and Methods
Line 127: remove “nematode” (it’s redundant)
- The required change has been addressed.
Line 132: Bursaphelenchus okinawaensis is not a plant-parasitic nematode; it’s a mycetophagous (fungus-feeding) nematode
- The required change has been addressed.
Results
Line 228: Scale bars are missing in Figure 2
- The required change has been addressed.
Lines 248-249: Italicize Heterodera latipons; also, even though it may be obvious, the meaning of different litters and their statistical significance should be clearly stated in the legend.
- The required change has been addressed.
Line 297: ClustalW Multiple Alignment algorithm, rather than program
- The required change has been addressed.
Discussion
Line 376: is it really worth abbreviating RNAi and PTGS? This is the only line in the entire manuscript where they are referred…
- The required change has been addressed.
Submission Date: 22 June 2024
Date of this review: 08 Jul 2024 15:54:21
Reviewer 2 Report
Comments and Suggestions for Authors
Dear author,
Manuscript Pathogens-3094812 provides useful information on the characterization of a population of the Mediterranean cereal cyst nematode (Heterodera latipons) collected from a barley field, in Jordan, during the warm and arid summer, in 2021. Additional data on the morphological and physiological changes of the cysts containing eggs with infective juveniles during their diapause are included in the paper. The findings obtained in this study enrich the knowledge on the biology and physiology of this nematode and can be published in the Journal Pathogens. Many sections of the manuscript should be clarified. A series of points in the manuscript that require the attention of the authors follows:
1) The title should reflect the morphological and molecular characterization of the population of H. latipons from Jordan. A suggested new title may be:’ Characterization of a population of the Mediterranean cereal cyst nematode (Heterodera latipons ), with notes on the morphological and biochemical changes of the cysts and enclosed eggs during their diapause.”
2) In the abstract, line 18, the authors may want to specify the life stage of the nematode that undergoes morphological and biochemical changes. The authors may want to provide information about the characterization of the studied population of H. latipons . They can write: “ In the current study, different morphological and biochemical changes of H. latipons cysts containing eggs with infective juveniles from a barley field, in Jordan, were studied during the summer 2021, at two sample dates. The first, at the harvest of the cereal crop (June, 2021), when the infective second-stage juveniles (J2s) were initiating diapause, and the second, before planting the sequent cereal crop (late October, 2021), when J2s were ending diapause. The studied population was characterized morphologically and molecularly and showed 98.4% molecular similarities to both JOD 176 from Jordan and Syrian “300” isolates of H. latipons. The obtained... “
3) In the abstract, line 22, the authors may want to emphasize that the number of the hatched J2s was the same at the two sample dates. They may want to write: “Slight change in the emergence time of J2s from cysts was observed without any difference in the number of hatched J2s. The results…. ”
4) In the abstract, line 28, the last sentence is too speculative. The authors may want to replace it with:” The outcomes of this study provide new helpful information that elucidates diapause in H. latipons and may be used for the implementation of new management strategies of the nematode.”
5) In the Introduction, line 44, the authors may want to provide information about the life cycle of Heterodera latipons. They may want to replace the sentence between lines 44 and 46 with this new sentence:” Its life cycle is initiated by second-stage juveniles (J2), which are motile in the soil and penetrate into the roots of a new cereal crop, where they become sedentary and develop into adult obese females after establishing permanent feeding sites. Obese females produce eggs that are retained inside their body, and at the end of their life, their cuticle hardens to form tough protective cysts, which remain attached to the roots or drop in the soil. Eggs complete embryonic development and contain newly molted J2s ready to hatch. The diapause of unhatched J2s is initiated by endogenous factors and terminated by exogenous stimuli. The phenomenon of diapause….”
6) In the Introduction, page 2, line 57, the authors should provide information about the population of cereal cyst nematode used in this study. The authors may want to write:” A population of a cereal cyst nematode, tentatively identified as H. latipons, was collected from a barley field on Jordan and used in this study. The correct identification of this population was necessary because several species of cyst-forming nematodes infest cereals in the Mediterranean basin. Their morphological characters are variable making their identification challenging. The objectives…”
7) In the introduction, page 2, line 57, the authors may want to add another objective. They can write: “The objectives of the present study were to : (i) characterize morphologically and molecularly the population of the cereal cyst nematode tentatively identified as H. latipons; from Jordan; (ii) determine the morphological and biochemical changes of cysts and enclosed J2s in eggs of this population during their diapause, and (iii) investigate whether genes responsible for diapause in C. elegans would have orthologs in this putative population of H. latipons.”
8) In Materials and Methods (MM) page 2, line 77, the authors exposed the cysts at temperatures of 8-10 C to break the dormancy of J2s in the cysts, without providing a reference about the source of this method to induce J2s hatching. They may want to write:” Low temperatures, 8-10 C, induce J2s hatching in studies conducted by Scholz (2001). The effect of these exogenous stimuli on J2s hatching was tested in the cysts collected at the two sampling dates. For this test, three replicates of 10 cysts each were placed into a 1.5 ml tube filled with 100 μl of SDW, and then maintained at 8 to 10°C in an incubator to verify the emergence of J2s from the cysts. " NOTE: This paragraph should be moved on line 93.
9) In MM, page 2, lines 80,81, the authors may want expands the information about the morphological analyses of cysts and J2s. The may want to write:’ For an accurate identification of the population from Jordan, comparisons of the morphological features of the conical portion of the posterior body of the cysts and morphometrics of 10 J2s and cysts were made with those of described populations of H. latipons and other species in the group avenae [2,25]. In addition, the findings of the morphological analysis were combined with those of the molecular characters. The obtained morphometrical data of J2s and cysts…”
10) In MM, page 2, lines 85,86, the authors may want to clarify the procedures used for the molecular analysis of the putative population of H.latipons from Jordan. They may want to write: “To confirm the results of the morphological analysis, molecular analyses of the cysts of the putative H. latipons population and other populations of this species from GeneBank were conducted using the 28rRNA gene sequences. The D2and D3 expansion segments of……[26]. Total…...”
11) In the results, page 4, lines 173-175, the authors may want to provide more details about the morphological identification of the putative populations of H. latipons. They may want to write:’ The morphological identification of the cereal cyst nematode population from Jordan as H. latipons was based on the comparisons of the morphological features and morphometrics of its J2s and cysts with those of H. latipons in the original description and other described populations [2] (Tables 1 and 2) (Supplementary figures S1and S2). In addition...”
12) In the Results, page 4, lines 178-172, the authors report the results of effect of low temperatures on J2s hatching. In this test the number of J2s hatching from the young (1–2-month-old) cysts (June,2021) was almost identical (600) to that (613) from older (4-month-old) cysts (October, 2021). These results differ from those reported by Scholtz (2001) who found more J2s hatching from old than young cysts. The authors may want to make some comments about these discrepancies in the discussion. Are the authors completely sure that the cysts collected from the field in June were all newly formed cysts and did not contain old cysts remained in the soil from the crop of the previous year? These old cysts may have provided more J2s hatching.
13) In Tables 1,2, page 4,5 lines 184-189, the authors should insert two additional columns with the morphological characters of the original population of H. latipons described by Franklin and those of Jordan isolate JOD, if available.
14) In the results page 7, line 225. The authors emphasize the importance of Figure 2 divided by subheadings. This sentence is redundant! The legend to figure 2 clearly explains the changes in the display of the sub-crystalline layer on the cysts at the two sampling dates.
15) In Supplementary Material pages 12, 13, the legends to figure S1 and S2 were corrected. The authors should replace the term fenestrae with semi-fenestrae.
16) In the References, page 12, In the reference 5, the authors should add the year 2001 and the pages 161.
Additional corrections are added to the text of the manuscript with sticky notes.

Comments on the Quality of English Language
no
Author Response
I and on behalf of all authors would like to thank this reviewer for his/her valuable suggestions and comments on the manuscript. We did the changes as kindly indicated.
Reviewer no.2
Comments and Suggestions for Authors
Dear author,
Manuscript Pathogens-3094812 provides useful information on the characterization of a population of the Mediterranean cereal cyst nematode (Heterodera latipons) collected from a barley field, in Jordan, during the warm and arid summer, in 2021. Additional data on the morphological and physiological changes of the cysts containing eggs with infective juveniles during their diapause are included in the paper. The findings obtained in this study enrich the knowledge on the biology and physiology of this nematode and can be published in the Journal Pathogens. Many sections of the manuscript should be clarified. A series of points in the manuscript that require the attention of the authors follows:
- The title should reflect the morphological and molecular characterization of the population of latiponsfrom Jordan. A suggested new title may be:’ Characterization of a population of the Mediterranean cereal cyst nematode (Heterodera latipons), with notes on the morphological and biochemical changes of the cysts and enclosed eggs during their diapause.”
- Thank you for your suggestion. We would like to point out that the Jordanian populations of latipons have already been characterized and published previously. In this paper, we focus on the morphological and biochemical changes during the diapause of this nematode. Therefore, we prefer and would appreciate it if the title remains as it is to reflect our objectives.
- In the abstract, line 18, the authors may want to specify the life stage of the nematode that undergoes morphological and biochemical changes. The authors may want to provide information about the characterization of the studied population of latipons. They can write: “In the current study, different morphological and biochemical changes of H. latipons cysts containing eggs with infective juveniles from a barley field, in Jordan, were studied during the summer 2021, at two sample dates. The first, at the harvest of the cereal crop (June, 2021), when the infective second-stage juveniles (J2s) were initiating diapause, and the second, before planting the sequent cereal crop (late October, 2021), when J2s were ending diapause. The studied population was characterized morphologically and molecularly and showed 98.4% molecular similarities to both JOD 176 from Jordan and Syrian “300” isolates of H. latipons. The obtained... “
- The required change has been addressed.
- In the abstract, line 22, the authors may want to emphasize that the number of the hatched J2s was the same at the two sample dates. They may want to write: “Slight change in the emergence time of J2s from cysts was observed without any difference in the number of hatched J2s. The results…. ”
- The required change has been addressed.
- In the abstract, line 28, the last sentence is too speculative. The authors may want to replace it with:” The outcomes of this study provide new helpful information that elucidates diapause in latipons and may be used for the implementation of new management strategies of the nematode.”
- The required change has been addressed.
- In the Introduction, line 44, the authors may want to provide information about the life cycle of Heterodera latipons. They may want to replace the sentence between lines 44 and 46 with this new sentence:” Its life cycle is initiated by second-stage juveniles (J2), which are motile in the soil and penetrate into the roots of a new cereal crop, where they become sedentary and develop into adult obese females after establishing permanent feeding sites. Obese females produce eggs that are retained inside their body, and at the end of their life, their cuticle hardens to form tough protective cysts, which remain attached to the roots or drop in the soil. Eggs complete embryonic development and contain newly molted J2s ready to hatch. The diapause of unhatched J2s is initiated by endogenous factors and terminated by exogenous stimuli. The phenomenon of diapause….”
- The required change has been addressed.
- In the Introduction, page 2, line 57, the authors should provide information about the population of cereal cyst nematode used in this study. The authors may want to write:” A population of a cereal cyst nematode, tentatively identified as latipons, was collected from a barley field on Jordan and used in this study. The correct identification of this population was necessary because several species of cyst-forming nematodes infest cereals in the Mediterranean basin. Their morphological characters are variable making their identification challenging. The objectives…”
- The required change has been addressed.
- In the introduction, page 2, line 57, the authors may want to add another objective. They can write: “The objectives of the present study were to : (i) characterize morphologically and molecularly the population of the cereal cyst nematode tentatively identified as latipons; from Jordan; (ii) determine the morphological and biochemical changes of cysts and enclosed J2s in eggs of this population during their diapause, and (iii) investigate whether genes responsible for diapause in C. eleganswould have orthologs in this putative population of H. latipons.”
- The required change has been addressed.
- In Materials and Methods (MM) page 2, line 77, the authors exposed the cysts at temperatures of 8-10 C to break the dormancy of J2s in the cysts, without providing a reference about the source of this method to induce J2s hatching. They may want to write:” Low temperatures, 8-10 C, induce J2s hatching in studies conducted by Scholz (2001). The effect of these exogenous stimuli on J2s hatching was tested in the cysts collected at the two sampling dates. For this test, three replicates of 10 cysts each were placed into a 1.5 ml tube filled with 100 μl of SDW, and then maintained at 8 to 10°C in an incubator to verify the emergence of J2s from the cysts. " NOTE: This paragraph should be moved on line 93.
- The required change has been addressed.
- In MM, page 2, lines 80,81, the authors may want expands the information about the morphological analyses of cysts and J2s. The may want to write:’ For an accurate identification of the population from Jordan, comparisons of the morphological features of the conical portion of the posterior body of the cysts and morphometrics of 10 J2s and cysts were made with those of described populations of latiponsand other species in the group avenae [2,25]. In addition, the findings of the morphological analysis were combined with those of the molecular characters. The obtained morphometrical data of J2s and cysts…”
- The required change has been addressed.
- In MM, page 2, lines 85,86, the authors may want to clarify the procedures used for the molecular analysis of the putative population of latiponsfrom Jordan. They may want to write: “To confirm the results of the morphological analysis, molecular analyses of the cysts of the putative H. latipons population and other populations of this species from GeneBank were conducted using the 28rRNA gene sequences. The D2and D3 expansion segments of……[26]. Total…...”
- The required change has been addressed.
- In the results, page 4, lines 173-175, the authors may want to provide more details about the morphological identification of the putative populations of latipons. They may want to write:’ The morphological identification of the cereal cyst nematode population from Jordan as H. latiponswas based on the comparisons of the morphological features and morphometrics of its J2s and cysts with those of H. latipons in the original description and other described populations [2] (Tables 1 and 2) (Supplementary figures S1and S2). In addition...”
- The required change has been addressed.
- In the Results, page 4, lines 178-172, the authors report the results of effect of low temperatures on J2s hatching. In this test the number of J2s hatching from the young (1–2-month-old) cysts (June,2021) was almost identical (600) to that (613) from older (4-month-old) cysts (October, 2021). These results differ from those reported by Scholtz (2001) who found more J2s hatching from old than young cysts. The authors may want to make some comments about these discrepancies in the discussion. Are the authors completely sure that the cysts collected from the field in June were all newly formed cysts and did not contain old cysts remained in the soil from the crop of the previous year? These old cysts may have provided more J2s hatching.
- The required change has been addressed.
- In Tables 1,2, page 4,5 lines 184-189, the authors should insert two additional columns with the morphological characters of the original population of latiponsdescribed by Franklin and those of Jordan isolate JOD, if available.
- Thank you for your suggestion. We have conducted a comparative analysis of the morphometrics of the latipons population from our study with other populations of H. latipons from Jordan, as well as with the original description by Franklin and other species within the Avenae group, using discriminant analysis. Based on this comparison, we believe that including the additional columns may not significantly contribute to the main findings of our study. Additionally, comprehensive tables with similar morphometric comparisons from Jordan have already been published in other studies.
- In the results page 7, line 225. The authors emphasize the importance of Figure 2 divided by subheadings. This sentence is redundant! The legend to figure 2 clearly explains the changes in the display of the sub-crystalline layer on the cysts at the two sampling dates.
- The required change has been addressed.
- In Supplementary Material pages 12, 13, the legends to figure S1 and S2 were corrected. The authors should replace the term fenestrae with semi-fenestrae.
- The required change has been addressed.
- In the References, page 12, In the reference 5, the authors should add the year 2001 and the pages 161.
- The required change has been addressed.
Additional corrections are added to the text of the manuscript with sticky notes. peer-review-38382697.v1.zip
Comments on the Quality of English Language: no
Submission Date: 22 June 2024
Date of this review: 19 Jul 2024 04:11:11